# COMPLEXITY OF TRAINING ReLU NEURAL NETWORK

## ABSTRACT

In this paper, we explore some basic questions on the complexity of training Neural networks with ReLU activation function. We show that it is NP-hard to train a two-hidden layer feedforward ReLU neural network. If dimension $d$ of the data is fixed then we show that there exists a polynomial time algorithm for the same training problem. We also show that if sufficient over-parameterization is provided in the first hidden layer of ReLU neural network then there is a polynomial time algorithm which finds weights such that output of the over-parameterized ReLU neural network matches with the output of the given data.

## 1 INTRODUCTION

Deep neural networks (DNNs) are functions computed on a graph parameterized by its edge weights. More formally, the graph corresponding to a DNN is defined by input and output dimensions $w_0, w_{k+1} \in \mathbb{Z}_+$, number of hidden layers $k \in \mathbb{Z}_+$, and by specifying a sequence of $k$ natural numbers $w_1, w_2, \ldots, w_k$ representing the number of nodes in each of the hidden $k$-layers. The function computed on the DNN graphs is:

$$f := a_{K+1} \circ \tau \circ a_K \circ \cdots \circ a_2 \circ \tau \circ a_1,$$

where $\circ$ is function composition, $\tau$ is a nonlinear function (applied componentwise) called as the activation function, and $a_i : \mathbb{R}^{w_{i-1}} \to \mathbb{R}^{w_i}$ are affine functions.

Given input and corresponding output data, the problem of training a DNN can be thought of as determining edge weights of directed layered graph for which output of the neural network matches the output data as closely as possible. Formally, given a set of input and output data $\{(x^i, y^i)\}_{i=1}^N$ where $(x^i, y^i) \in \mathbb{R}^{w_0} \times \mathbb{R}^{w_{k+1}}$, and a loss function $l : \mathbb{R}^{w_{k+1}} \times \mathbb{R}^{w_{k+1}} \to \mathbb{R}_+$ (for example, $l$ can be a norm) the task is to determine the weights that define the affine function $a_i$'s such that

$$\sum_{i=1}^N l(f(x^i), y^i) \tag{1}$$

is minimized.

Some commonly studied activation functions are: threshold function, sigmoid function and ReLU function. ReLU is one of the important activation functions used widely in applications and literature. However, the problem of complexity of training multi-layer fully-connected ReLU neural network remained open. This is where we add our contributions. Before formally stating the results, we take a look at current state-of-the-art in the literature.

**Complexity of training DNNs with threshold activation function** The threshold (sign) function is given by

$$\text{sgn}(x) := \begin{cases} 1 & \text{if } x > 0 \\ -1 & \text{if } x < 0 \end{cases}.$$

Neural network with threshold activation function is also called as binary neural network in modern machine learning literature. It was shown by Blum et al. Blum & Rivest (1988) that problem of training a simple two layer neural network with two nodes in the first layer and one node in the second layer while using threshold activation function at all the nodes is NP-complete. The problem

turns out to be equivalent to separation by two hyperplanes which was shown to be NP-complete by Megiddo (1988). There are other hardness results such as crypto hardness for intersection of k-hyperplanes which apply to binary neural networks Shalev-Shwartz & Ben-David (2014); Klivans & Sherstov (2009). In Shalev-Shwartz & Ben-David (2014) it is shown that even the problem of training a binary neural network with 3 or more nodes in first hidden layer and 1 node in second hidden layer is NP-hard.

**DNNs with rectified linear unit (ReLU) activation function**    Theoretical worst case results presented above, along with limited empirical successes lead to DNN's going out of favor by late 1990s. However, in recent time, DNNs became popular again by advent of first-order gradient based heuristics for training. This success started with the work of Hinton et al. (2006) which gave empirical evidence that if DNNs are initialized properly then we can find good solutions in reasonable runtime. This work was soon followed by series of early successes of deep learning in natural language processing Collobert & Weston (2008), speech recognition Mohamed et al. (2012) and visual object classification Krizhevsky et al. (2012). It was empirically shown by Zhang et al. (2016) that a sufficiently over-parameterized neural network can be trained to global optimality.

These gradient-based heuristics are not useful for binary neural networks as there is no gradient information. Even networks with sigmoid activation function fell out of favor because gradient information is not valuable when input values are large. The popular neural network architecture uses *ReLU activations* on which gradient based heuristics are useful. Fomally, the ReLU function is given by: $[x]_+ := \max(x, 0)$.

**Related literature**    As discussed before, most hardness results so far are for binary neural networks Blum & Rivest (1988); Klivans & Sherstov (2009); Shalev-Shwartz & Ben-David (2014). There are also limited results for ReLU that we discuss next: Recently, Livni et al. (2014) examined ReLU activations from the point of view that, a shifted ReLU subtracted from another ReLU yields an approximation to threshold function, so such a class of ReLU network should be as hard as binary neural network. Similar results are shown by DasGupta et al. (1994). In both these papers, in order to model "a shifted ReLU subtracted from another ReLU", the Neural network studied is not a fully connected network. More specifically, in the underlying graph of such a neural network, each node in the second hidden layer is connected to exactly one distinct node in the first hidden layer, weight of the connecting edge is set to −1 with the addition of some positive bias term. Figure 1 shows the difference between ReLU network studied by Livni et al. (2014); DasGupta et al. (1994) and fully connected ReLU network. Clearly, the architecture described in Livni et al. (2014); DasGupta et al. (1994) artificially restrict the form of the affine functions in order to prove NP-hardness. In particular, it requires connecting hidden layer matrix to be a square diagonal matrix. Due to this restriction, it was unclear whether allowing full matrix to be non-zero would make problem easy (more parameters hence higher power to neural network function) or hard (more parameters so more things to decide).

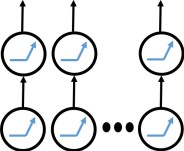

(a) ReLU network studied in Livni et al. (2014); DasGupta et al. (1994)

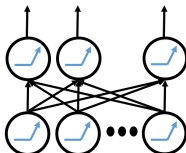

(b) Fully connected ReLU

Figure 1: Difference between ReLU model studied in Livni et al. (2014); DasGupta et al. (1994) and typical fully connected counterpart

Another interesting line of research in understanding the hardness of training ReLU neural networks assumes that data is coming from some distribution. More recent work in this direction include Shamir (2016) which shows a smooth family of functions for which gradient of squared error function is not informative while training neural network over Gaussian input distribution. Song et al. (2017) consider Statistical Query (SQ) framework (which contains SGD algorithms) and show that

there exists a class of special functions generated by single hidden layer neural network for which learning will require exponential number of queries (i.e. sample gradient evaluations) for data coming from product measure of real valued log-concave distribution. These are interesting studies in their own right and generally consider hardness with respect to the algorithms that use stochastic gradient queries to the population objective functions. In comparison, we consider the framework of NP-hardness which takes into account complete class of polynomial time algorithms, generally assumes that data is given and looks at corresponding empirical objective.

Recently, Arora et al. (2016) showed that a single hidden layer ReLU network can be trained in polynomial time when dimension of input, $w_0$, is constant.

Based on the above discussion, we see that the complexity status of training the multi-layer fully-connected ReLU neural network remains open. Given the importance of the ReLU NN, this is an important question. In this paper, we take the first steps in resolving this question.

**Main Contributions**

- NP-hardness: We show that the training problem for a simple two hidden layer fully connected NN which has two nodes in the first layer, one node in the second layer and ReLU activation function at all nodes is NP-hard. Underlying graph of this network is exactly the same as that in Blum et al. Blum & Rivest (1988) but all activation functions are ReLU instead of threshold function. Techniques used in the proof are different from earlier work in literature because there is no combinatorial interpretation to ReLU as opposed to threshold function.

- Polynomial-time solvable cases: We present two cases where the training problem with ReLU activation function can be solved in polynomial-time. The first case is when the dimension of the input is fixed. The second case is when we have a network where the number of nodes in the first layer is equal to the number of input data points (highly over-parameterized neural network). This second result leads to interesting open questions that we discuss later.

## 2 NOTATION AND DEFINITIONS

We use the following standard set notation $[n] := \{1, \dots, n\}$. The letter $d$ generally denotes the dimension of input data, the output data is 1 dimensional and $N$ denotes the number of data-points.

The main training problem of interest for the paper corresponds to a neural network with 3 nodes. The underlying graph is a layered directed graph with two layers. The first layer contains two nodes and the second layer contains one node. The network is fully connected feedforward network. One can write the function corresponding to this neural network as follows:

$$F(x) = \theta\big[w_0 + w_1\big[a_1(x)\big]_+ + w_2\big[a_2(x)\big]_+\big]_+, \tag{2}$$

where $a_i : \mathbb{R}^d \to \mathbb{R}$ for $i \in \{1, 2\}$ are real valued affine functions, and $w_0, w_1, w_2 \in \mathbb{R}$. The output

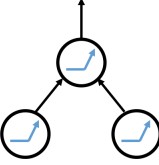

Figure 2: (2,1)-ReLU Neural Network. Also called 2-ReLU NN after dropping '1'. Here ReLU function is presented in each node to specify the type of activation function at the output of each node.

of the two affine maps $a_1, a_2$ are the inputs to the two ReLU nodes in first hidden layer of network. The weights $\{w_0, w_1, w_2\}$ denote affine map for ReLU node in second layer. The coefficient $\theta \in \mathbb{R}$ is the linear map of the output layer. Henceforth, we refer to the network defined in (2) as (2,1)-ReLU Neural Network(NN). As its name suggests, it has 2 ReLU nodes in first layer and 1 ReLU

node in second layer. Figure 2 shows $(2, 1)$-ReLU NN.
Note that

$$w[ax + b]_+ \equiv \text{sgn}(w)[|w|(ax + b)]_+ = \text{sgn}(w)[\tilde{a}x + \tilde{b}],$$

so without loss of generality we will assume $w_1, w_2 \in \{-1, 1\}$.

We will refer to $(k, j)$-ReLU NN as generalization of $(2, 1)$-ReLU NN where there are $k$ ReLU nodes in first layer and $j$ ReLU nodes in second layer. Moreover, output of $(k, j)$-ReLU NN lies in $\mathbb{R}^j$.

If there is only one node in the second layer, we will often drop the $'1'$ and refer it as a 2-ReLU NN or k-ReLU NN depending on whether there are 2 or $k$ nodes in the first layer.

**Definition 2.1 (Decision-version of training problem)** *Given a set of training data $(x^i, y^i) \in \mathbb{R}^d \times \{1, 0\}$ for $i \in S$, does there exist edge weights so that the resulting function $F$ satisfies $F(x^i) = y^i$ for $i \in S$.*

The decision version of the training problem in Definition 2.1 is asking if it is possible to find edge weights to obtain zero loss function value in the expression (1), assuming $l$ is a norm i.e. $l(a, b) = 0$ iff $a = b$.

# 3 MAIN RESULTS

**Theorem 3.1** *It is NP-hard to solve the training problem for 2-ReLU NN.*

The proof of Theorem 3.1 is obtained by reducing the 2-Affine Separability Problem to the training problem of 2-ReLU NN. Details of this reduction and the proof is presented in Section 4. Corollary of Theorem above is as follows:

**Corollary 3.2** *Training problem of (2,j)-ReLU NN is NP hard, for all $j \geq 1$.*

Megiddo (1988) shows that the separability with fixed number of hyperplanes (generalization of 2-affine separability problem) can be solved in polynomial-time in fixed dimension. Therefore 2-affine separability problem can be solved in polynomial time given dimension is constant. Based on the reduction used to prove Theorem 3.1 , a natural question to ask is "Can one solve the training problem of 2-ReLU NN problem in polynomial time under the same assumption?". We answer this question in the affirmative.

**Theorem 3.3** *Under the assumption that dimension of input, $d$, is a constant, there exist a poly(N) solution to the training problem of 2-ReLU neural network, where $N$ is number of data-points.*

A proof of this Theorem is presented in Appendix A.6. This theorem suggests that hardness of learning is due to high dimension and as long as $d$ is small, we can find reasonably good algorithms (for practical purposes) for large values of $N$.

We also study this problem under over-parameterization. Structural understanding of 2-ReLU NN yields an easy algorithm to solve training problem for N-ReLU neural network over N data points.

**Theorem 3.4** *Given data, $\{x^i, y^i\}_{i \in [N]}$, then the training problem for N-ReLU NN has a poly(N,d) randomized algorithm, where $N$ is the number of data-points and $d$ is the dimension of input.*

A proof of this Theorem is presented in Appendix A.7.

# 4 TRAINING 2-RELU NN IS NP-HARD

In this section we give details about the NP-hardness reduction for the training problem of 2-ReLU NN. We begin with the formal definition of 2-Affine Separability Problem.

**Definition 4.1 (2-Affine Separability Problem)** *Given a set of points $\{x^i\}_{i \in [N]} \in \mathbb{R}^d$ and a partition of $[N]$ into two sets: $S_1, S_0$, (i.e. $S_1 \cap S_0 = \emptyset$, $S_1 \cup S_0 = [N]$) decide whether there exist two hyperplanes $H_1 = \{x : \alpha_1^T x + \beta_1 = 0\}$ and $H_2 = \{x : \alpha_2^T x + \beta_2 = 0\}(\alpha_1, \alpha_2 \in \mathbb{R}^d, \beta_1, \beta_2 \in \mathbb{R})$ that separate the set of points in the following fashion:*

  *i For each point $x^i$ such that $i \in S_1$, both $\alpha_1^T x^i + \beta_1 > 0$ and $\alpha_2^T x^i + \beta_2 > 0$.*

*ii For each point $x^i$ such that $i \in S_0$, either $\alpha_1^T x^i + \beta_1 < 0$ or $\alpha_2^T x^i + \beta_2 < 0$.*

The problem 2-affine separability is NP-complete Megiddo (1988). Note the difference between conditions i and ii above. First one is an "AND" statement and second is an "OR" statement. Geometrically, solving 2-affine separability problem means that finding two affine hyperplanes $\{\alpha_1, \beta_1\}$ and $\{\alpha_2, \beta_2\}$ such that all points in set $S_1$ lie in one quadrant formed by two hyperplanes and all points in set $S_0$ lie outside that quadrant. Due to this geometric intuition, the problem is called separation by 2-hyperplanes or 2-affine separability. We will construct a polynomial reduction from this NP-complete problem to training 2-ReLU NN, which will prove that training 2-ReLU NN is NP-hard.

**Remark** [Variants of 2-affine separability]:
Note here that some sources also define 2-affine separability problem with minor difference. In particular, the change is that strict inequalities, '>', in Definition 4.1.i are diluted to inequalities, '≥'. In fact, these two problems are equivalent in the sense that there is solution for the first problem if and only if there is a solution for the second problem. Solution for the first problem implies solution for the second problem trivially. Suppose there is a solution for the second problem, that implies there exist $\{\alpha_1, \beta_1\}$ and $\{\alpha_2, \beta_2\}$ such that for all $i \in S_0$ we have either $\alpha_1^T x^i + \beta_1 < 0$ or $\alpha_2^T x^i + \beta_2 < 0$. This implies $\epsilon := \min_{i \in S_0} \max\{-\alpha_1 x^i - \beta_1, -\alpha_2 x^i - \beta_2\} > 0$. So if we shift both planes by $\frac{1}{2}\epsilon$ i.e. $\beta_i \leftarrow \beta_i + \frac{1}{2}\epsilon$ then this is a solution to the first problem.

**Assumption: $\mathbf{0} \in S_1$** (Here $\mathbf{0} \in \mathbb{R}^d$ is a vector of zeros.) Suppose we are given a generic instance of 2-affine separability problem with data-points $\{x^i\}_{i \in [N]}$ from $\mathbb{R}^d$ and partition $S_1/S_0$ of set $[N]$. Since the answer of 2-affine separability instance is invariant under coordinate translation, we shift the origin to any $x^i$ for $i \in S_1$, and therefore assume that the origin belongs to $S_1$ henceforth.

## 4.1  REDUCTION

Now we create a particular instance for 2-ReLU NN problem from general instance of 2-affine separability. We add two new dimensions to each data-point $x^i$. We also create a label, $y^i$, for each data-point. Moreover, we add a constant number of extra points to the training problem. Exact details are as follows:

Consider training set $\{(x^i, 0, 0), y^i\}_{i \in [N]}$ where $y^i = \begin{cases} 1 & \text{if } i \in S_1 \\ 0 & \text{if } i \in S_0 \end{cases}$.

Add additional 12 data points to the above training set as follows:
$\{p_1 \equiv \{(\mathbf{0}, 1, 1), 1\}, p_2 \equiv \{(\mathbf{0}, 2, 1), 1\}, p_3 \equiv \{(\mathbf{0}, 1, 2), 1\}, p_4 \equiv \{(\mathbf{0}, 2, 2), 1\},$
$p_5 \equiv \{(\mathbf{0}, 1, -1), 0\}, p_6 \equiv \{(\mathbf{0}, 2, -1), 0\}, p_7 \equiv \{(\mathbf{0}, 3, -1), 0\},$
$p_8 \equiv \{(\mathbf{0}, -1, 1), 0\}, p_9 \equiv \{(\mathbf{0}, -1, 2), 0\}, p_{10} \equiv \{(\mathbf{0}, -1, 3), 0\},$
$p_{11} \equiv \{(\mathbf{0}, -1, 0), 0\}, p_{12} \equiv \{(\mathbf{0}, 0, -1), 0\}\}.$
Lets call the set of additional data points with label 1 as $T_1$ and additional data points with label 0 as $T_0$. These additional data points (we refer to these points as the "gadget points") are of fixed size.

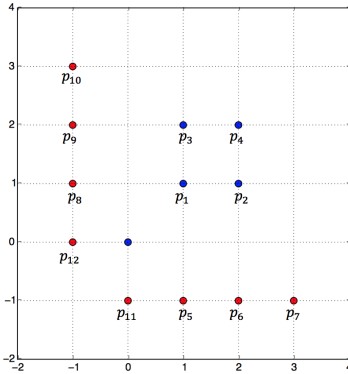

Figure 3: Gadget: Blue points represent set $T_1$ and red points represent set $T_0$

So this is a polynomial time reduction.

Figure 3 shows the gadget points. Note that origin is added to the gadget because there exists $i \in S_1$ such that $x^i = \mathbf{0}$. Hence training set has the data-point $\{(\mathbf{0}, 0, 0), 1\}$.

Lets call the training problem of fitting 2-ReLU NN to this data as (**P**). Now what remains is to show that general instance of 2-affine separability has a solution if and only if constructed instance of 2-ReLU NN has a solution. We will first show the forward direction of reduction. This is also the easier direction.

**Lemma 4.1** *If 2-affine separability problem has a solution then problem (**P**) has a solution.*

The proof of Lemma 4.1 can be found in appendix section A.1.
To prove reverse direction we need to show that if a set of weights solve the training problem (**P**) then we can generate a solution to the 2-affine separation problem. To understand the approach we take to prove this direction, we introduce the notion of "hard-sorting". Hard-sorting is formally defined below, and its significance is stated in Lemma 4.4.

**Definition 4.2 (Hard-sorting)** *We say that a set of points $\{\pi^i\}_{i \in S}$, partitioned into two sets $\Pi_1, \Pi_2$ can be hard-sorted with respect to $\Pi_1$ if there exist two affine transformations $l_1, l_2$ and scalars $w_1, w_2, c$ such that either one of the following two conditions are satisfied:*

*1.* $w_1 \big[ l_1(\pi) \big]_+ + w_2 \big[ l_2(\pi) \big]_+ \begin{cases} = c & \text{for all } \pi \in \Pi_1 \\ > c & \text{for all } \pi \in \Pi_2 \end{cases}$

*2.* $w_1 \big[ l_1(\pi) \big]_+ + w_2 \big[ l_2(\pi) \big]_+ \begin{cases} = c & \text{for all } \pi \in \Pi_1 \\ < c & \text{for all } \pi \in \Pi_2 \end{cases}$

Being able to hard-sort implies that after passing data through two nodes of the first hidden layer, the scalar input to the second hidden layer node must have a separation of the data-points in $\Pi_1$ and the data-points in $\Pi_2$, moreover, scalar input for all data points in $\Pi_1$ must be a constant.

**Remark 4.2** *Hard-sorting property is invariant under sign change of both $w_1, w_2$.*

**Remark 4.3** *Let $\overline{\Pi}_1 \subseteq \Pi_1$ and $\overline{\Pi}_2 \subset \Pi_2$. Then hard-sorting of $\Pi_1 \cup \Pi_2$ with respect to $\Pi_1 \Rightarrow$ hard-sorting of $\overline{\Pi}_1 \cup \overline{\Pi}_2$ with respect to $\overline{\Pi}_1$.*

It is not difficult to see that hard-sorting implies (**P**) has a solution. We show that hard-sorting is also required for solving training problem. This is formally stated in lemma below.

**Lemma 4.4** *The 2-ReLU NN training problem (**P**) has a solution if and only if data-points $S_1 \cup T_1 \cup S_0 \cup T_0$ are hard-sorted with respect to $S_1 \cup T_1$*

The proof of Lemma 4.4 can be found in appendix section A.2 .
Figure 4 below explains geometric interpretation of Lemma 4.4

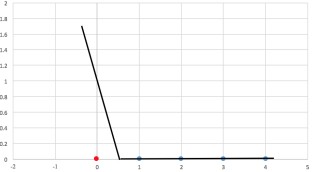 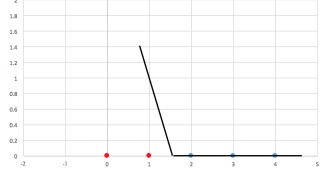 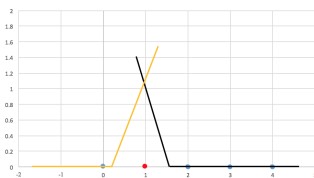

(a) Input is hard-sorted. This can give a perfect fit.

(b) Since there are two red points so input is not hard-sorted. This cannot give a perfect fit.

(c) Since blue points lies on different side of red points so input is not hard-sorted. This cannot give a perfect fit.

Figure 4: X-axis in figures above is output of the first layer of 2-ReLU NN i.e. $w_1\big[l_1(\pi)\big]_+ + w_2\big[l_2(\pi)\big]_+$. Y-axis is the output of second hidden layer node. Since output of first hidden layer goes to input of second hidden layer, we are essentially trying to fit ReLU node of second hidden layer. In particular, red and blue dots represent output of first hidden layer on data points with label 1 and 0 respectively. In fig (a) we see that hard-sorted input can be classified as $0/1$ by a ReLU function. In fig (b) and (c) we see that input which is not hard-sorted can not be classified exactly as $0/1$ by a ReLU function.

In the rest of the proof we will argue that the only way to solve the training problem for 2-ReLU NN (**P**) or equivalently hard-sort data-points is to find two affine function $a_1, a_2$ such that i) $a_1(x) \leq 0$ and $a_2(x) \leq 0$ for all $x \in S_1 \cup T_1$ and ii) $a_1(x) > 0$ or $a_2(x) > 0$ for all $x \in S_0 \cup T_0$. If such a solution exists then there exists a solution to 2-affine separability problem after dropping coefficients of last two dimensions of affine functions $-a_1$ and $-a_2$.

We will first show that we can hard-sort the gadget points only under the properties of $a_1$ and $a_2$ mentioned above. This implies that a solution to (**P**) which hard-sorts all points (including the gadget points) must have same properties of $a_1$ and $a_2$. This follows from counter-positive to Remark 4.3 i.e. if subset of data-points can not be hard-sorted then all data-points can not be hard-sorted. So henceforth, we will focus on the gadget data-points (or the last two dimensions of the data).

### 4.1.1 Gadget Points and Hard-Sorting

For lemma provided below, we assume $a_1, a_2 : \mathbb{R}^2 \to \mathbb{R}$ are affine functions and $\mathbf{0} \in \mathbb{R}^2$. They can be thought of as projection of original $a_i : \mathbb{R}^{d+2} \to \mathbb{R}$ and $\mathbf{0} \in \mathbb{R}^{d+2}$ to last two dimension which are relevant for gadget data points added according to set $T_1 \cup T_0$.

**Lemma 4.5** *Suppose affine functions $a_1, a_2 : \mathbb{R}^2 \to \mathbb{R}$ satisfy hard-sorting of the data-points $T_1 \cup T_0 \cup \{\mathbf{0}\}$ with respect to $T_1 \cup \{\mathbf{0}\}$ then all points $x \in T_1$ **must** satisfy $a_1(x) \leq 0$ and $a_2(x) \leq 0$ with at least one inequality being a strict inequality.*

First observe that if $a_1$ and $a_2$ satisfy hard-sorting, then neither of them can be a constant function. This is trivially true if both of them are constant. If one of them is constant then data needs to be linearly separable which is not the case for gadget data-points $T_1 \cup T_0 \cup \{\mathbf{0}\}$. So we will assume that both of them are affine functions with non-zero normal vector. Now, the proof of Lemma 4.5 is divided into the following sequence of propositions:

**Proposition 4.6** *Suppose there exists affine functions $a_1, a_2 : \mathbb{R}^2 \to \mathbb{R}$, such that $a_1(x) = 0$ and $a_2(x) = 0$ are parallel lines. Moreover, assume that magnitude of the normal to these lines is equal i.e. $\|\nabla a_1\| = \|\nabla a_2\| \neq 0$. Then such $a_1, a_2$ can not satisfy hard-sorting of data points $T_1 \cup T_0 \cup \{\mathbf{0}\}$ with respect to $T_1 \cup \{\mathbf{0}\}$*

**Remark 4.7** *A key corollary of Proposition 4.6 is that if $a_1, a_2$ satisfy hard-sorting of gadget data points $T_1 \cup T_0 \cup \{\mathbf{0}\}$ with respect to $T_1 \cup \{\mathbf{0}\}$ then set $L := \{x | w_1 a_1(x) + w_2 a_2(x) = c\}$ is a line for all $c \in \mathbb{R}$. Henceforth, in the proofs of subsequent propositions, we will refer $L$ as $w_1 a_1 + w_2 a_2 = c$ hiding the input variable, $x$, for ease of notation.*

**Proposition 4.8** *Suppose there exists affine functions $a_1, a_2 : \mathbb{R}^2 \to \mathbb{R}$, such that $a_1(x) \geq 0$ and $a_2(x) \geq 0$ for **any 3** points, $x \in T_1$, then such functions can not satisfy hard-sorting of $T_1 \cup T_0 \cup \{\mathbf{0}\}$ with respect to $T_1 \cup \{\mathbf{0}\}$.*

**Proposition 4.9** *Suppose there exists affine functions $a_1, a_2 : \mathbb{R}^2 \to \mathbb{R}$, such that $a_1(x) \geq 0$ and $a_2(x) \geq 0$ for **exactly** 2 points, $x \in T_1$, then such functions can not satisfy hard-sorting of $T_1 \cup T_0 \cup \{\mathbf{0}\}$ with respect to $T_1 \cup \{\mathbf{0}\}$.*

**Proposition 4.10** *Suppose there exists affine functions $a_1, a_2 : \mathbb{R}^2 \to \mathbb{R}$, such that $a_1(x) \geq 0$ and $a_2(x) \geq 0$ for **exactly** 1 point, $x \in T_1$, then such functions can not satisfy hard-sorting of $T_1 \cup T_0 \cup \{\mathbf{0}\}$ with respect to $T_1 \cup \{\mathbf{0}\}$.*

From Propositions 4.8 , 4.9 and 4.10 , we conclude that no point $x \in T_1$ can be on non-negative side of both affine functions when those affine functions are required to satisfy hard-sorting for the gadget points. Now we will show that any single point $x \in T_1$ can **not** be on the positive side of even one of the affine functions.

**Proposition 4.11** *Suppose there exists affine functions $a_1, a_2 : \mathbb{R}^2 \to \mathbb{R}$, such that $a_1(x) > 0$ or $a_2(x) > 0$ for any $x \in T_1$, the such functions can not satisfy hard-sorting of $T_1 \cup T_0$ with respect to $T_1$*

Note that Propositions 4.8 , 4.9 , 4.10 and 4.11 imply that given affine functions $a_1, a_2$ hard-sorting $T_1 \cup T_0 \cup \{\mathbf{0}\}$ with respect to $T_1 \cup \{\mathbf{0}\}$, all points $x \in T_1$ must satisfy $a_1(x) \leq 0$ and $a_2(x) \leq 0$ with at least one of them being strict inequality. It is clear that each $x \in T_1$ must satisfy inequalities $a_1(x) \leq 0, a_2(x) \leq 0$. At least one of these inequalities has to be strictly negative otherwise we have a contradiction to Proposition 4.10.This proves Lemma 4.5.

Now we show one more simple lemma which is critical in proving the final result.

**Lemma 4.12** *Affine functions $a_1, a_2$ and weights $w_1, w_2$ satisfy hard-sorting of $T_1 \cup T_0 \cup \{\mathbf{0}\}$ with respect to $T_1 \cup \{\mathbf{0}\}$ then parity of weights **must** be same i.e. $w_1 = w_2 = 1$ or $w_1 = w_2 = -1$.*

In the next section, we show that this result on gadget data-points gives us solution to the original 2-affine separability problem.

### 4.1.2   FROM GADGET DATA TO COMPLETE DATA

Note that if there is a solution to problem (**P**), say $a_1, a_2 \in \mathbb{R}^{d+2}$ and $w_1, w_2, w_0, \theta \in \mathbb{R}$, then by Lemma 4.5 , Lemma 4.12 and counterpositive of Remark 4.3 we have

1. Parity of $w_1, w_2$ is same.
2. $w_1 \big[a_1(x)\big]_+ + w_2 \big[a_2(x)\big]_+ = 0$ for all $x \in S_1 \cup T_1$ due to requirement of hard-sorting.

Since parity of $w_1, w_2$ is same so 2 above is satisfied if $a_1(x) \leq 0$ and $a_2(x) \leq 0$ for all $x \in S_1 \cup T_1$. Moreover, we require $a_1(x) > 0$ or $a_2(x) > 0$ for all $x \in S_0 \cup T_0$ because otherwise we will violate hard-sorting. Now as discussed earlier, $-a_1, -a_2$ after ignoring coefficients of last two dimensions will yield solution to 2-affine separability problem. So we have the following lemma.

**Lemma 4.13** *If there is a solution to the problem (**P**), then there is a solution to corresponding 2-Affine separability problem.*

**Theorem 4.14** *Training problem for 2-ReLU NN is NP-hard.*

**Proof.** Using Lemma 4.1 and Lemma 4.13, we conclude the proof. □

Immediate corollary of Theorem 4.14 is as follows:

**Corollary 4.15** *Training problem of (2,j)-ReLU NN is NP hard.*

## 5   DISCUSSION

We showed that the problem of training 2-ReLU NN is NP-hard. Given the importance of ReLU activation function in neural networks, in our opinion, this result resolves a significant gap in understanding complexity class of the problem at hand. On the other hand, we show that the problem of training $N$-ReLU NN is in P. So a natural research direction is to understand the complexity status when input layer has more than 2 nodes and strictly less than $N$ nodes. A particularly interesting question in that direction is to generalize the gadget we used for 2-ReLU NN to the case of k-ReLU NN.

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

# A    PROOFS OF AUXILIARY RESULTS

In this appendix, we provide proof of all auxiliary results.

## A.1    PROOF OF LEMMA 4.1

Suppose $(\alpha_1, \beta_1)$ and $(\alpha_2, \beta_2)$ are solution satisfying condition for 2-affine separability. Note that there is a data-point $\mathbf{0} \in S_1$ so we obtain $\beta_1, \beta_2 > 0$. Without loss of generality we can assume $\beta_1 = \beta_2 = 0.5$. This is due to the fact that scaling the original solution by any positive scalar yields a valid solution. Define $\epsilon := \min_{i \in S_0} \left\{ \left[ -\alpha_1^T x^i - \beta_1 \right]_+ + \left[ -\alpha_2^T x^i - \beta_2 \right]_+ \right\}$. Note that since we have a valid solution to 2-affine separability, we must have $\epsilon > 0$ because sum of two non-negative quantities where at least one is positive must be positive. Define $\eta := \min\{\frac{1}{2}\epsilon, \frac{1}{4}\}$. By definition $\eta > 0$ and hence $\eta^{-1}$ is defined. Now 2-ReLU neural network function can be written as follows:

$$f(x, y, z) = \frac{1}{\eta}\left[ -\left[ -\alpha_1^T x - \beta_1 - y \right]_+ - \left[ -\alpha_2^T x - \beta_2 - z \right]_+ + \eta \right]_+.$$

We claim that 2-ReLU NN with weights assigned as above solves the training problem (**P**).

1. For $x \in S_1$, we have

$$f(x, 0, 0) = \frac{1}{\eta}\left[ -\left[ -\alpha_1^T x - \beta_1 \right]_+ - \left[ -\alpha_2^T x - \beta_2 \right]_+ + \eta \right]_+ = \frac{1}{\eta}[0 + 0 + \eta] = 1.$$

2. For $x \in S_0$, we have

$$f(x, 0, 0) = \frac{1}{\eta}\left[ -\left[ -\alpha_1^T x - \beta_1 \right]_+ - \left[ -\alpha_2^T x - \beta_2 \right]_+ + \eta \right]_+.$$

   Now note that $\left[ -\alpha_1^T x - \beta_1 \right]_+ + \left[ -\alpha_2^T x - \beta_2 \right]_+ \geq \epsilon, \forall\, x \in S_0$. So inner term of above expression is less than or equal to $-\epsilon + \eta$. Since $\eta \leq \frac{1}{2}\epsilon$, the inner term is negative. Hence for all $x \in S_0$, $f(x, 0, 0) = 0$.

3. For $x = (\mathbf{0}, l, m) \in T_1$, we have

$$f(\mathbf{0}, l, m) = \frac{1}{\eta}\left[ -\left[ -\beta_1 - l \right]_+ - \left[ -\beta_2 - m \right]_+ + \eta \right]_+.$$

   Note that since $\beta_1 = \beta_2 = 1/2$ and $l, m \in \{1, 2\}$ so the two ReLU terms inside are both zero for all $x \in T_1$. Hence $f(x) = 1$.

4. For $x = (\mathbf{0}, l, m) \in T_0$, we have

$$f(\mathbf{0}, l, m) = \frac{1}{\eta}\left[ -\left[ -\beta_1 - l \right]_+ - \left[ -\beta_2 - m \right]_+ + \eta \right]_+.$$

   Note that since $\beta_1 = \beta_2 = 1/2$ and either $l$ or $m$ equals $-1$, we obtain that either one of the ReLU terms equals 1/2. So inner term is less than or equal to $-1/2 + \eta$, but $\eta \leq 1/4$ therefore $f(x) = 0$ for all $x \in T_0$.

This proves existence of weights solving the training problem (**P**).

## A.2    PROOF OF LEMMA 4.4

Suppose points are hard-sorted as required by lemma with condition 1 of hard sorting. Then define $\epsilon := \min_{x \in S_0 \cup \overline{S}_0} w_1\left[ l_1(x) \right]_+ + w_2\left[ l_2(x) \right]_+ - c$. By definition $\epsilon > 0$. Then it is easy to check that neural network $f(x) = \frac{2}{\epsilon}\left[ -w_1\left[ l_1(x) \right]_+ - w_2\left[ l_2(x) \right]_+ + c + \epsilon/2 \right]_+$ solves training problem. Similar arguments hold when points can be hard-sorted with condition 2 of hard-sorting.

Now we assume that points can not be hard-sorted and conclude that there does not exist weight assignment solving training problem of 2-ReLU NN. Since the points cannot be hard-sorted so there does not exist any $l_1, l_2, w_1, w_2, c$ satisfying either condition 1 or condition 2. This implies for all possible weights we either have

a) $w_1 \big[ l_1(x) \big]_+ + w_2 \big[ l_2(x) \big]_+$ is not constant for all $x \in S_1 \cup \overline{S}_1$ or

b) If $w_1 \big[ l_1(x) \big]_+ + w_2 \big[ l_2(x) \big]_+ = c$ for all $x \in S_1 \cup \overline{S}_1$ and some constant $c$, then same expression evaluated on $x \in S_0 \cup \overline{S}_0$ is not strictly on same side of $c$.

If we choose $l_1, l_2, w_1, w_2, c$ such that a) happens, then such weights will not solve training problem as their output of 2-ReLU NN for points $p \in S_1 \cup \overline{S}_1$ will be at least two distinct numbers which is an undesirable outcome. Specifically, we want $\theta \big[ w_0 + w_1 \big[ l_1(x) \big]_+ + w_2 \big[ l_2(x) \big]_+ \big]_+$ to evaluate to 1 for all $x \in S_1 \cup \overline{S}_1$ so we must have $\theta > 0$ and $w_0 + w_1 \big[ l_1(x) \big]_+ + w_2 \big[ l_2(x) \big]_+$ take a constant positive value for all $x \in S_1 \cup \overline{S}_1$. Hence $w_1 \big[ l_1(x) \big]_+ + w_2 \big[ l_2(x) \big]_+$ must be a constant for all $x \in S_1 \cup \overline{S}_1$. This requirement is violated in case a).

If we choose $l_1, l_2, w_1, w_2, c$ such that b) happens, then we can set $w_0, \theta$ such that $F(x) = \theta \big[ w_1 \big[ l_1(x) \big]_+ + w_2 \big[ l_2(x) \big]_+ + w_0 \big]_+$, $w_0 + c > 0$ and $\theta = \frac{1}{w_0 + c}$. Since not all $x \in S_0 \cup \overline{S}_0$ are strictly on one side, we conclude there exist $x' \in S_0 \cup \overline{S}_0$ such that $w_1 \big[ l_1(x') \big]_+ + w_2 \big[ l_2(x') \big]_+ = c' \geq c$ hence $F(x') := \theta \big[ w_1 \big[ l_1(x') \big]_+ + w_2 \big[ l_2(x') \big]_+ + w_0 \big]_+ \geq 1$ which is an undesirable outcome for a point with label 0.

Since all choices of $l_1, l_2, w_1, w_2, c$ satisfy either a) or b) so we conclude that there does not exist weights solving training problem of 2-ReLU NN.

## A.3 Proof of Proposition 4.6

We assume that magnitude of the normal to these lines is equal i.e. $\|\nabla a_1\| = \|\nabla a_2\| \neq 0$. Note that we have two possible situations here: $a_1, a_2$ satisfy 1)$a_1(x) + a_2(x) = c, \forall\, x \in \mathbb{R}$ when normals point in opposite directions and 2) $a_1(x) - a_2(x) = c, \forall x \in \mathbb{R}$ when normals point in same direction. (Here $c \in \mathbb{R}$ is a constant). We will consider both these cases separately and show that expression $w_1 \big[ a_1 \big]_+ + w_2 \big[ a_2 \big]_+$ can not hard-sort as required irrespective of the parity of weights $w_1, w_2$. Due to Remark 4.2, we just need to check for case $\{w_1, w_2\} = \{1, 1\}$ and $\{w_1, w_2\} = \{1, -1\}$. More specifically, $\{w_1, w_2\} = \{-1, -1\}$ yields a hard-sorting solution iff there exists a hard-sorting solution for $\{w_1, w_2\} = \{1, 1\}$. Equivalent argument can be made about the case $\{w_1, w_2\} = \{-1, 1\}$ and $\{w_1, w_2\} = \{1, -1\}$

**Case 1:** Normals point in opposite direction. Let $a_1 + a_2 = c$. Suppose $c \geq 0$. Then it can be verified that

$$\big[ a_1 \big]_+(x) + \big[ a_2 \big]_+(x) = \begin{cases} c & \text{if } c \geq a_1(x) \geq 0 \\ a_1(x) & \text{if } a_1(x) \geq c \\ c - a_1(x) & \text{if } a_1(x) \leq 0. \end{cases}$$

By hard-sorting requirement, we need all five points in $T_1 \cup \{\mathbf{0}\}$ should be in set $\{x : a_1(x) \in [0, c]\}$ and all points in $T_0$ should not be in this set. Now observe that if $c = 0$, then the set $\{x : a_1(x) = 0\}$ is one dimensional, and therefore can not contain all the points of $T_1 \cup \{\mathbf{0}\}$. Hence we must have $c > 0$. It can be seen that this is impossible to achieve by two parallel lines $a_1 = 0$ and $a_1 = c$ for the given set up of data points $T_0 \cup T_1 \cup \{\mathbf{0}\}$. Similarly when $c < 0$, then it can be verified that

$$\big[ a_1 \big]_+(x) + \big[ a_2 \big]_+(x) = \begin{cases} 0 & \text{if } c \leq a_1(x) \leq 0 \\ a_1(x) & \text{if } a_1(x) \geq 0 \\ c - a_1(x) & \text{if } a_1(x) \leq c \end{cases}$$

Again, for hard-sorting, as in the previous case, we need all five points in $T_1 \cup \{\mathbf{0}\}$ should be in set $\{x : a_1(x) \in [c, 0]\}$ and all points in $T_0$ should not be in this set which can not be achieved.

Now consider case where parity of $w_i$'s is different. When $c \geq 0$, it can be verified that

$$\big[ a_1 \big]_+(x) - \big[ a_2 \big]_+(x) = \begin{cases} a_1(x) - c & \text{if } a_1(x) \leq 0 \\ 2a_1(x) - c & \text{if } 0 \leq a_1(x) \leq c \\ a_1(x) & \text{if } a_1(x) \geq c \end{cases}$$

It is clear that all five points in $T_1 \cup \mathbf{0}$ can not be on line $a_1(x) = \delta$ for any constant $\delta$. So this type of function can not hard-sort given points.

When $c < 0$. Then it can be verified that

$$[a_1]_+(x) - [a_2]_+(x) = \begin{cases} a_1(x) & \text{if } a_1(x) \geq 0 \\ 0 & \text{if } 0 \geq a_1(x) \geq c \\ a_1(x) - c & \text{if } a_1(x) \leq c \end{cases}$$

For hard-sorting with this type of function, we again need all points in $T_1 \cup \{\mathbf{0}\}$ to be in set $\{x : a_1(x) \in [c, 0]\}$ while all remaining points to be outside this set which is not possible to be achieved by any affine function $a_1$.

**Case 2:** $a_1 - a_2 = c$. Suppose $c \geq 0$. Then it can be verified that

$$[a_1]_+(x) - [a_2]_+(x) = \begin{cases} a_1(x) & \text{if } c \geq a_1(x) \geq 0 \\ c & \text{if } a_1(x) \geq c \\ 0 & \text{if } a_1(x) \leq 0 \end{cases}$$

If $c = 0$ then $[a_1]_+(x) - [a_2]_+(x) = 0$ for all $x \in \mathbb{R}^2$. So this can not hard-sort data. Hence for hard-sorting we definitely need $c > 0$. Moreover, we need either 1) $T_1 \cup \{\mathbf{0}\} \subset \{x : a_1(x) \leq 0\}$ and $T_0 \subset \{x : a_1(x) > 0\}$ or 2) $T_1 \cup \{\mathbf{0}\} \subset \{x : a_1(x) \geq c\}$ and $T_0 \subset \{x : a_1(x) < c\}$. So essentially the points in $T_1 \cup T_0 \cup \{\mathbf{0}\}$ must be separable by a line. This is not possible.

Note that when $c < 0$, one can write $a_2 - a_1 = -c$ and write similar functional form for $[a_2]_+ - [a_1]_+$.

Now consider case when parity of weights $w_i$'s is same. Again suppose $c \geq 0$. Then one can verify that

$$[a_1]_+(x) + [a_2]_+(x) = \begin{cases} 0 & \text{if } a_1(x) \leq 0 \\ a_1(x) & \text{if } c \geq a_1(x) \geq 0 \\ 2a_1(x) - c & \text{if } a_1(x) \geq c \end{cases}$$

Clearly, for hard-sorting we need strict separation by the line $a_1(x) = 0$ which is not possible. When $c < 0$ then we can write $a_2 - a_1 = -c$ and we will need strict separation at line $a_2(x) = 0$ which is again not possible.

Since in both cases, none of the parity combinations were able to achieve hard-sorting $T_1 \cup T_0 \cup \{\mathbf{0}\}$ w.r.t. $T_1 \cup \{\mathbf{0}\}$, so we conclude the proof.

### A.3.1 Proof of Proposition 4.6

Note that we have two possible situations here: $a_1, a_2$ satisfy 1)$a_1(x) + a_2(x) = c, \forall \, x \in \mathbb{R}$ when normals point in opposite directions and 2) $a_1(x) - a_2(x) = c, \forall x \in \mathbb{R}$ when normals point in same direction. (Here $c \in \mathbb{R}$ is a constant). We will consider both these cases separately and show that expression $w_1[a_1]_+ + w_2[a_2]_+$ can not hard-sort as required irrespective of the parity of weights $w_1, w_2$. Due to Remark 4.2, we just need to check for case $\{w_1, w_2\} = \{1, 1\}$ and $\{w_1, w_2\} = \{1, -1\}$. More specifically, $\{w_1, w_2\} = \{-1, -1\}$ yields a hard-sorting solution iff there exists a hard-sorting solution for $\{w_1, w_2\} = \{1, 1\}$. Equivalent argument can be made about the case $\{w_1, w_2\} = \{-1, 1\}$ and $\{w_1, w_2\} = \{1, -1\}$

**Case 1:** Normals point in opposite direction. Let $a_1 + a_2 = c$. Suppose $c \geq 0$. Then it can be verified that

$$[a_1]_+(x) + [a_2]_+(x) = \begin{cases} c & \text{if } c \geq a_1(x) \geq 0 \\ a_1(x) & \text{if } a_1(x) \geq c \\ c - a_1(x) & \text{if } a_1(x) \leq 0. \end{cases}$$

By hard-sorting requirement, we need all five points in $\overline{S}_1 \cup \{\mathbf{0}\}$ should be in set $\{x : a_1(x) \in [0, c]\}$ and all points in $\overline{S}_0$ should not be in this set. Now observe that if $c = 0$, then the set $\{x : a_1(x) = 0\}$ is one dimensional, and therefore can not contain all the points of $\overline{S}_1 \cup \{\mathbf{0}\}$. Hence we must have $c > 0$. It can be seen that this is impossible to achieve by two parallel lines $a_1 = 0$ and $a_1 = c$ for the given set up of data points $\overline{S}_0 \cup \overline{S}_1 \cup \{\mathbf{0}\}$.

Similarly when $c < 0$, then it can be verified that

$$[a_1]_+(x) + [a_2]_+(x) = \begin{cases} 0 & \text{if } c \leq a_1(x) \leq 0 \\ a_1(x) & \text{if } a_1(x) \geq 0 \\ c - a_1(x) & \text{if } a_1(x) \leq c \end{cases}$$

Again, for hard-sorting, as in the previous case, we need all five points in $\overline{S}_1 \cup \{\mathbf{0}\}$ should be in set $\{x : a_1(x) \in [c, 0]\}$ and all points in $\overline{S}_0$ should not be in this set which can not be achieved. Now consider case where parity of $w_i$'s is different. When $c \geq 0$, it can be verified that

$$
\left[a_1\right]_+(x) - \left[a_2\right]_+(x) = \begin{cases} a_1(x) - c & \text{if } a_1(x) \leq 0 \\ 2a_1(x) - c & \text{if } 0 \leq a_1(x) \leq c \\ a_1(x) & \text{if } a_1(x) \geq c \end{cases}
$$

It is clear that all five points in $\overline{S}_1 \cup \mathbf{0}$ can not be on line $a_1(x) = \delta$ for any constant $\delta$. So this type of function can not hard-sort given points.
When $c < 0$. Then it can be verified that

$$
\left[a_1\right]_+(x) - \left[a_2\right]_+(x) = \begin{cases} a_1(x) & \text{if } a_1(x) \geq 0 \\ 0 & \text{if } 0 \geq a_1(x) \geq c \\ a_1(x) - c & \text{if } a_1(x) \leq c \end{cases}
$$

For hard-sorting with this type of function, we again need all points in $\overline{S}_1 \cup \{\mathbf{0}\}$ to be in set $\{x : a_1(x) \in [c, 0]\}$ while all remaining points to be outside this set which is not possible to be achieved by any affine function $a_1$.
**Case 2:** $a_1 - a_2 = c$. Suppose $c \geq 0$. Then it can be verified that

$$
\left[a_1\right]_+(x) - \left[a_2\right]_+(x) = \begin{cases} a_1(x) & \text{if } c \geq a_1(x) \geq 0 \\ c & \text{if } a_1(x) \geq c \\ 0 & \text{if } a_1(x) \leq 0 \end{cases}
$$

If $c = 0$ then $\left[a_1\right]_+(x) - \left[a_2\right]_+(x) = 0$ for all $x \in \mathbb{R}^2$. So this can not hard-sort data. Hence for hard-sorting we definitely need $c > 0$. Moreover, we need either 1) $\overline{S}_1 \cup \{\mathbf{0}\} \subset \{x : a_1(x) \leq 0\}$ and $\overline{S}_0 \subset \{x : a_1(x) > 0\}$ or 2) $\overline{S}_1 \cup \{\mathbf{0}\} \subset \{x : a_1(x) \geq c\}$ and $\overline{S}_0 \subset \{x : a_1(x) < c\}$. So essentially the points in $\overline{S}_1 \cup \overline{S}_0 \cup \{\mathbf{0}\}$ must be separable by a line. This is not possible.
Note that when $c < 0$, one can write $a_2 - a_1 = -c$ and write similar functional form for $\left[a_2\right]_+ - \left[a_1\right]_+$.
Now consider case when parity of weights $w_i$'s is same. Again suppose $c \geq 0$. Then one can verify that

$$
\left[a_1\right]_+(x) + \left[a_2\right]_+(x) = \begin{cases} 0 & \text{if } a_1(x) \leq 0 \\ a_1(x) & \text{if } c \geq a_1(x) \geq 0 \\ 2a_1(x) - c & \text{if } a_1(x) \geq c \end{cases}
$$

Clearly, for hard-sorting we need strict separation by the line $a_1(x) = 0$ which is not possible. When $c < 0$ then we can write $a_2 - a_1 = -c$ and we will need strict separation at line $a_2(x) = 0$ which is again not possible.
Since in both cases, none of the parity combinations were able to achieve hard-sorting $\overline{S}_1 \cup \overline{S}_0 \cup \{\mathbf{0}\}$ w.r.t. $\overline{S}_1 \cup \{\mathbf{0}\}$, so we conclude the proof.

### A.3.2 PROOF OF PROPOSITION 4.8

From Proposition 4.6 we obtain that $w_1 a_1 + w_2 a_2 = c$ is a line irrespective of parity of $w_1, w_2$ and any value of $c$. Define set $S_+ := \{x : a_1(x) \geq 0 \text{ and } a_2(x) \geq 0\}$. Then $\{x \in S_+ : w_1 \left[a_1\right]_+ + w_2 \left[a_1\right]_+(x) = c\}$ is a one dimensional set. Now any 3 points in $\overline{S}_1$ which are given to be in $S_+$ can not be on a single line (because they are not collinear). So we get that any 3 points in $\overline{S}_1$ can not be in $S_+$ while condition for hard-sorting holds.

### A.3.3 PROOF OF PROPOSITION 4.9

By Proposition 4.6, we know that $w_1 a_1 + w_2 a_2 = c$ is a line irrespective of parity of weights $w_1, w_2$ and value of constant $c$.
**Case 1**: Parity of $w_1, w_2$ is same.
Suppose exactly $p_3, p_4$ are two points in $\overline{S}_1$ which satisfy required conditions. Suppose $\left[a_1\right]_+(x) + \left[a_2\right]_+(x) = c$ for all $x \in \overline{S}_1$. Then $a_1(p_i) + a_2(p_i) = c, i = 3, 4$. Note that $c \geq 0$ and $0 \leq a_1(p_i) \leq$

$c, i = 3, 4$.

We first claim that $c > 0$. Assume by contradiction, $c = 0$. In this case $a_1(p_3) = a_1(p_4) = a_2(p_3) = a_2(p_4) = 0$. Since $p_3, p_4$ are distinct points, this implies that $\{x : a_1(x) = 0\}$ and $\{x : a_2(x) = 0\}$ are the same line and this line passes through points $p_3$ and $p_4$. Since $p_9$ is in affine hull of $p_3$ and $p_4$, we therefore have that $a_1(p_9) = a_2(p_9) = 0$. This clearly violates condition of hard-sorting. Thus $c > 0$.

The observation that $c > 0$ together with Proposition 4.8 gives us that $p_1, p_2$ must be separated from $p_3, p_4$ by **exactly** one of the lines $a_1 = 0$ and $a_2 = 0$. (At least one line because of Proposition 4.8 i.e. we can not allow $p_1$ to be in $S_+ := \{x : a_1(x) \geq 0 \text{ and } a_2(x) \geq 0\}$ along with $p_3, p_4$. At most one line because of observation $c > 0$, so $[a_1]_+(p_1) + [a_2]_+(p_1)$ must equals $c$ which is strictly positive quantity. Similar arguments can be made for $p_2$).

In essence, we get exactly two sub-cases:

1. Line $a_1(x) = 0$ separates only $p_1$ and $a_2(x) = 0$ separates only $p_2$. Equivalently, $a_1(p_1) < 0$ and $a_2(p_2) < 0$.
   Since $a_1(p_1) < 0 \leq a_1(p_3)$ so we conclude $a_1(p_2) < a_1(p_4)$. But as noted before, we have $a_1(p_4) \leq c$ which implies $a_1(p_2) < c$. So $[a_1]_+(p_2) + [a_2]_+(p_2) = a_1(p_2) < c$, contradiction to condition of hard-sorting.

2. $a_1 = 0$ separates both $p_1, p_2$ and $a_2 = 0$ separates none of $p_1, p_2$. Equivalently, $a_1(p_1) < 0, a_1(p_2) < 0$.
   $[a_1]_+(p_i) + [a_2]_+(p_i) = a_2(p_i), i = 1, 2$. So $a_2(p_i) = c, i = 1, 2$. Hence $a_2$ is constant for all points on line passing through $p_1, p_2$. Hence $a_2(p_3) = a_2(p_4)$. Hence $a_1(p_3) = a_1(p_4)$. Again, since $p_3$ and $p_4$ are distinct points and $p_9$ is on the affine hull of $p_3, p_4$, this implies that $a_1(p_3) = a_1(p_9)$ and $a_2(p_3) = a_2(p_9)$, and hence $p_3, p_4$ and $p_9$ will have the same output, a contradiction to condition of hard-sorting.

Similar arguments can be made about pairs $\{p_1, p_2\}, \{p_1, p_3\}$ and $\{p_2, p_4\}$.

Suppose $p_2, p_3 \in S_+$. Then $\{x : a_1(x) + a_2(x) = c\}$ line passes through $p_2, p_3$. Note that this line strictly separates $p_1, p_4$. So we may assume $a_1(p_4) + a_2(p_4) > c$. Also by Proposition 4.8, $p_4$ must be separated from $p_2, p_3$ by at least one of the lines $a_1 = 0, a_2 = 0$. So we may assume $a_1(p_4) < 0 \Rightarrow a_2(p_4) > c \Rightarrow [a_1]_+(p_4) + [a_2]_+(p_4) > c$. Contradiction to condition of hard-sorting.

Similar argument can be made about pair $\{p_1, p_4\}$.

**Case 2** Parity of $w_1, w_2$ is different.

Suppose $[a_1]_+ - [a_2]_+ = c$. Suppose $c > 0$. (We will argue $c = 0$ separately. Arguments for $c < 0$ will go through in similar fashion by interchanging $a_1, a_2$ and replacing $c$ by $-c$).

Suppose $p_3, p_4 \in S_+ := \{x : a_1(x) \geq 0, a_2(x) \geq 0\}$. It is clear that for hard-sorting $a_1(x) \geq c > 0, \forall x \in \{p_1, p_2\}$. So by Proposition 4.8, we must have $a_2(x) < 0, \forall x \in \{p_1, p_2\}$. This implies $a_1(x) = c$ line passes through $p_1, p_2$. So $a_1(p_3) = a_1(p_4) \Rightarrow a_2(p_3) = a_2(p_4)$. So lines $a_1 = 0, a_2 = 0$ are parallel to line passing through $p_3, p_4$. Hence $[a_1]_+(p_9) - [a_2]_+(p_9) = c$, a contradiction to condition of hard-sorting.

Suppose $c = 0$. Then $a_1(x) - a_2(x) = 0$ line passes through $p_3, p_4$. Consider $q$, point of intersection of lines $a_1 = 0, a_2 = 0$ (These lines can't be parallel. Otherwise line $a_1 - a_2 = 0$ will be parallel to line $a_1 = 0$ and $a_2 = 0$. We know that $a_1 - a_2 = 0$ passes through $p_3, p_4$. So $a_1 = 0$ and $a_2 = 0$ will be parallel to line passing through $p_3, p_4$. This implies $p_9$ will have same output as $p_3, p_4$. This is contradiction to condition of hard-sorting. So a unique point, $q$, exists).

Point $q$ trivially lies on line $a_1 - a_2 = 0$ or equivalently on the line passing through $p_3, p_4$. We claim that $p_9 \notin S_+$. Assume by contradiction that $p_9 \in S_+$. We know that $a_1 - a_2$ passes through $p_3, p_4$ and $p_9$ is in affine combination of $p_3, p_4$ so $a_1(p_9) - a_2(p_9) = 0$. This implies $[a_1]_+(p_9) - [a_1]_+(p_9) = 0$, a contradiction to condition of hard-sorting. So $p_9$ must not be in $S_+$.

Since $p_3, p_4 \in S_+$ and $p_9 \notin S_+$ so we obtain that $q$ must lie on line segment $p_9 p_3$. Hence lines $a_1 = 0, a_2 = 0$ must separate $p_3, p_9$. So $a_1(p_9) \leq 0, a_2(p_9) \leq 0 \Rightarrow [a_1]_+(p_9) - [a_2]_+(p_9) = 0$. Since $c = 0$, this is contradiction to condition of hard-sorting.

Similar arguments can be made about pairs $\{p_1, p_2\}, \{p_1, p_3\}$ and $\{p_2, p_4\}$.

Suppose $p_2, p_3 \in S_+$. Then by Proposition 4.8, either $a_1(p_1) < 0$ or $a_2(p_1) < 0$. Without loss of generality, we may assume $a_1(p_1) < 0$. Since $a_1(p_3) \geq 0$ we get $a_1(p_3) > a_1(p_1) \Rightarrow a_1(p_4) > a_1(p_2) \geq 0$. Since $a_1(p_4) > 0$, by Proposition 4.8, we conclude $a_2(p_4) < 0 \leq a_2(p_3) \Rightarrow a_2(p_1) >$

$a_2(p_2) \geq 0$. Now $\left[a_1\right]_+(p_1) - \left[a_2\right]_+(p_1) = -a_2(p_1) < 0$ and $\left[a_1\right]_+(p_4) - \left[a_2\right]_+(p_4) = a_1(p_4) > 0$. So we get a contradiction.

Similar argument can be made about pair $\{p_1, p_4\}$.

### A.3.4 PROOF OF PROPOSITION 4.10

By Proposition 4.6, we know that $w_1 a_1 + w_2 a_2 = c$ is a line irrespective of parity of weights $w_1, w_2$ and value of constant $c$.

**Case 1:** Parity of $w_1, w_2$ is same.

Suppose $\left[a_1\right]_+(x) + \left[a_2\right]_+(x) = c$ for all points $x \in \overline{S}_1 \cup \{\mathbf{0}\}$. We assume $c > 0$. (We will argue about $c = 0$ separately). Suppose $p_1 \in S_+ := \{x : a_1(x) \geq 0, a_2(x) \geq 0\}$. Then by Propositions 4.8, 4.9, we conclude that all points in $x \in \overline{S}_1 \setminus \{p_1\}$ must satisfy **exactly** one of the following inequalities: $a_1(x) \geq 0, a_2(x) \geq 0$. (At most one inequality because of Propositions 4.8, 4.8 i.e. we can not allow $p_i, i = 2, 3, 4$ to be in $S_+ := \{x : a_1(x) \geq 0 \text{ and } a_2(x) \geq 0\}$ along with $p_1$. At least one inequality because of observation $c > 0$, so $\left[a_1\right]_+(p_1) + \left[a_2\right]_+(p_1)$ must equals $c$ which is strictly positive quantity.)

Hence there are two distinct cases to consider:

1. Any two of $\overline{S}_1 \setminus \{p_1\}$ satisfy $a_1(x) \geq 0, a_2(x) < 0$. Remaining one satisfy $a_1(x) < 0, a_2(x) \geq 0$. So $a_2(x) \geq 0$ is satisfied by only two points in $\overline{S}_1$ one of which is $p_1$. By arrangement of points in set $\overline{S}_1$ it is clear that either
   1) $a_2(p_i) \geq 0, i = 1, 2 \Rightarrow a_1(p_i) \geq 0, i = 1, 3, 4$ or
   2) $a_2(p_i) \geq 0, i = 1, 3 \Rightarrow a_1(p_1) \geq 0, i = 1, 2, 4$.
   These two are rotationally equivalent cases so we will only prove for case 1). Case 2) will have similar proof.
   For hard-sorting, $\left[a_1\right]_+(p_3) + \left[a_2\right]_+(p_3) = \left[a_1\right]_+(p_4) + \left[a_2\right]_+(p_4)$. Since $a_2(p_i) < 0, i = 3, 4$ so we get $a_1(p_3) = a_1(p_4)$. This implies $a_1(p_1) = a_1(p_2)$. Contradiction to assumptions of case 1.

2. All points in $\overline{S}_1 \setminus \{p_1\}$ satisfy $a_1(x) \geq 0, a_2(x) < 0$. Then for satisfying conditions of hard-sorting, we must have $\left[a_1\right]_+(p_i) + \left[a_2\right]_+(p_i) = a_1(p_i) = c, i = 2, 3, 4$. Clearly $p_2, p_3, p_4$ are not collinear so this is impossible.

Similar arguments can be made about $p_2, p_3, p_4$ when $c > 0$.

Suppose $c = 0$ and $p_1 \in S_+$. Then line $a_1 = 0, a_2 = 0$ passes through $p_1$. Moreover $\left[a_1\right]_+(p_i) + \left[a_2\right]_+(p_i) = 0, i = 2, 3, 4$ implies $a_1(p_i) \leq 0, a_2(p_i) \leq 0, i = 2, 3, 4$. Now note that $p_4$ can not lie on the line $\{x : a_1(x) = 0\}$ because otherwise $\{x : a_1(x) = 0\}$ will be line passing through $p_1, p_4$ hence strictly separate $p_2, p_3$. So either one of $p_2, p_3$ will satisfy $a_1(x) > 0$ which is contradiction to the fact that $\left[a_1\right]_+(x) + \left[a_2\right]_+(x) = 0$ for all $x \in \{p_2, p_3\}$. Similarly $p_4$ can not lie on $\{x : a_2(x) = 0\}$. Hence $a_1(p_4) < 0, a_2(p_4) < 0$ and $a_1(p_1) = 0, a_2(p_1) = 0$. Since $\mathbf{0}, p_1, p_4$ are collinear, we get that $a_1(\mathbf{0}) > 0, a_2(\mathbf{0}) > 0$. But then $\left[a_1\right]_+ + \left[a_2\right]_+(\mathbf{0}) > 0$. This contradicts condition of hard-sorting.

Suppose $p_2 \in S_+$. Then line $a_1 = 0, a_2 = 0$ passes through $p_2$. Moreover $\left[a_1\right]_+ + \left[a_2\right]_+(p_i) = 0, i = 1, 3, 4$ implies $a_1(p_1) \leq 0, a_2(p_1) \leq 0$. So by colinearity of $p_2, p_1, p_8$ we conclude that $a_1(p_8) \leq 0, a_2(p_8) \leq 0$. Then $\left[a_1\right]_+(p_8) + \left[a_2\right]_+(p_8) = 0$ which is contradiction to condition fo hard-sorting.

For $p_3$ or $p_4$ while $c = 0$: arguments similar to that of $p_2$ can be made.

**Case 2:** Parity of $w_1, w_2$ is different.

Suppose $p_1 \in S_+$ and $\left[a_1\right]_+(x) - \left[a_2\right]_+(x) = c$ for all $x \in \overline{S}_1 \cup \{\mathbf{0}\}$. Assume $c > 0$. Proof for $c < 0$ will follow from this case with $a_1, a_2$ exchanged and $c$ replaced with $-c$. (We will argue for $c = 0$ separately).

Since $c > 0$, we conclude $a_1(p_i) \geq c > 0, i = 2, 3, 4$. By statement of Proposition it implies $a_2(p_i) < 0, i = 2, 3, 4$. Then $\left[a_1\right]_+(p_i) + \left[a_2\right]_+(p_i) = a_1(p_i), i = 2, 3, 4$. Since $p_2, p_3, p_4$ are not collinear points, so we get contradiction to condition of hard-sorting.

Similar arguments will work for $p_2, p_3, p_4$.

Suppose $c = 0$ and $p_1 \in S_+$. If $a_1(p_2) > 0$ then $a_2(p_2) > 0$ but only $p_1 \in S_+$ so $a_1(p_2) \leq 0, a_2(p_2) \leq 0$. Similarly $a_1(p_i) \leq 0, a_2(p_i) \leq 0, i = 3, 4$. By Proposition 4.9, we have that either

$a_1(p_4) < 0$ or $a_2(p_4) < 0$. Without loss of generality, we assume $a_1(p_4) < 0$. Since $p_4, p_1, \mathbf{0}$ are collinear so $a_1(\mathbf{0}) > 0$. Since $c = 0$ and $[a_1]_+(\mathbf{0}) - [a_2]_+(\mathbf{0}) = c$ so we conclude that $a_2(\mathbf{0}) > 0$. So $a_1 - a_2 = 0$ line passes through $\mathbf{0}, p_1$. Note that by collinearity of $p_3, p_1, p_5$ and $a_i(p_3) \leq 0, a_i(p_1) = 0$ for $i = 1, 2$ we obtain $a_1(p_5) \geq 0, a_2(p_5) \geq 0$. Similarly by collinearity of $p_2, p_1, p_8$, we obtain $a_1(p_8) \geq 0, a_2(p_8) \geq 0$. So $\mathbf{0}, p_1, p_5, p_8 \in S_+$. Moreover $[a_i]_+(x) = a_i(x), \forall\, x \in S_+$. So we get $[a_1]_+(x) - [a_2]_+(x) = a_1(x) - a_2(x)$ for all $x \in \{\mathbf{0}, p_1, p_5, p_8\}$. Since $a_1 - a_2 = 0$ lines passes through $\mathbf{0}, p_1$, it separates $p_5, p_8$. So we get contradiction.

Suppose $p_2 \in S_+$ and $c = 0$. So we have $a_1(p_1) \leq 0, a_2(p_1) \leq 0$. By collinearity of $p_2, p_1, p_8$ we get $a_1(p_8) \leq 0, a_2(p_8) \leq 0 \Rightarrow [a_1]_+ - [a_2]_+(p_8) = 0$ which is a contradiction to assumption of hard-sorting.

Similar arguments can be made about $p_3, p_4$.

### A.3.5 PROOF OF PROPOSITION 4.11

**Case 1:** Parity of $w_1, w_2$ is same.
Suppose $p_1 \in S_+ := \{x : a_1(x) > 0\}$. Let $T_+ := \{x : a_2(x) > 0\}$. Then by Propositions 4.8, 4.9, 4.10, we have $a_2(p_1) < 0$. So $[a_1]_+(p_1) + [a_2]_+(p_1) = c > 0$. For hard-sorting, each $x \in \overline{S}_1$ must be on strictly on positive side of **exactly** one of the lines. So we conclude that $S_+, T_+$ is a partition of $\overline{S}_1$. Note that if there are any three points in set $\overline{S}_1 \cap S_+$ then $a_1(x)$ will have at least two distinct values for points in set $\overline{S}_1 \cap S_+$ because of non-collinearity. For points $x \in S_+ \cap \overline{S}_1$, we have $[a_1]_+(x) + [a_2]_+(x) = a_1(x)$. So we have contradiction to condition of hard-sorting. Similarly there can not be any three points in $T_+ \cap \overline{S}_1$. So both $S_+, T_+$ must contain exactly two points from $\overline{S}_1$. Clearly $\{p_2, p_3\}$ or $\{p_1, p_4\}$ can not be in $S_+$ or $T_+$. So the partition has to be either
1) $\{p_1, p_2\}, \{p_3, p_4\}$ or
2) $\{p_1, p_3\}, \{p_2, p_4\}$.
If 1) then lines $a_1 = 0, a_2 = 0$ are parallel to line passing through $p_1, p_2$. So $[a_1]_+ + [a_2]_+(p_i)$ will be same for $i = 1, 2, 8$ which is a contradiction to condition of hard-sorting. Similar contradiction can be found in case of 2).

**Case 2:** Parity of $w_1, w_2$ is different.
If $a_1(p_1) > 0$ then $a_2(p_1) < 0$ (By Propositions 4.8, 4.9, 4.10). Then $[a_1]_+(x) - [a_2]_+(x) = a_1(p_1)$ for all $x \in \overline{S}_1$. This implies $a_1(x) > 0$ for all $x \in \overline{S}_1 \Rightarrow a_2(x) < 0$ for all $x \in \overline{S}_1$. Then $[a_1]_+(x) - [a_2]_+(x) = a_1(x)$ for all $x \in \overline{S}_1$. But then $a_1(x)$ can not be a constant number for $x \in \overline{S}_1$ which yields a contradiction.

### A.4 PROOF OF LEMMA 4.12

Lemma 4.5 yields that any hard-sorting $a_1, a_2$ must satisfy $a_1(x) \leq 0, a_2(x) \leq 0$ for all $x \in T_1 \cup \{\mathbf{0}\}$.

Suppose parity of $w_1, w_2$ is different. Since $a_1$ and $a_2$ satisfy hard-sorting of gadget so we have $[a_1(x)]_+ - [a_2(x)]_+ = c, \forall\, x \in T_1$. Due to Lemma 4.5, $c = 0$. Then to fulfill hard-sorting condition, we need $[a_1(x)]_+ - [a_2(x)]_+ > 0 \forall\, x \in T_0$. (Case for "$[a_1(x)]_+ - [a_2(x)]_+ < 0$" will have same proof with all $a_1$ exachanged by $a_2$ in next 3 lines.)

This implies $a_1(x) > 0$ for all $x \in T_0$. In particular, we note that $a_1(p_7) > 0, a_1(p_{10}) > 0$. Moreover, $a_1(p_1) \leq 0$ as $p_1 \in T_1$. But $p_7, p_1, p_{10}$ are collinear points so $p_1$ is not separable from $p_7, p_{10}$ by an affine function. So we get a contradiction to the assumption that parity of weights $w_1, w_2$ is different.

### A.5 PROOF OF COROLLARY 4.15

The reduction is similar except the labels need to be changed from $\mathbb{R}$ to $\mathbb{R}^j$. Simply add $j - 1$ zeros to original output labels. Now output of $j - 1$ nodes is 0 for all data-points so these are redudnnant. In particular, for $k \in [j]$, every $k$-th node in the second layer is connected to 2 nodes in the first layer by distict edges whose weights are parameterized by $w_{k,1}, w_{k,2}$ and bias weight $w_{k,0}$. We can set $w_{k,1} = w_{k,2} = -1$ and $w_{k,0} = 0$ for all $k \in [j] \setminus \{1\}$. This yields the output 0 at all nodes $k \in [j] \setminus \{1\}$, irrespective of the affine functions $a_1, a_2$ in the first layer. Now, first node

satisfied to global optimality will yield solution $a_1, a_2, w_{1,1}, w_{1,2}, w_{1,0}$. By the reduction, we know that $-a_1, -a_2$ after ignoring last two co-ordinates yield solution to 2-affine separability problem.

## A.6 PROOF OF THEOREM 3.3

We will use result by Edelsbrunner et al. (1986). This work shows that given a set of points $\{x^i\}_{i \in N}$ in $\mathbb{R}^d$ we can enumerate all possible partition created by linear separators in $O(N^d)$ time.

Suppose we partition $[N]$ into $Q_1/\overline{Q}_1$ and $Q_2/\overline{Q}_2$ such that $a_1(x) = 0$ separates $Q_1/\overline{Q}_1$ and $a_2(x) = 0$ separates $Q_2/\overline{Q}_2$.

We define $T_1 = Q_1 \cap Q_2$, $T_2 = \overline{Q}_1 \cap Q_2$, $T_3 = Q_1 \cap \overline{Q}_2$ and $T_4 = \overline{Q}_1 \cap \overline{Q}_2$. Let $z = (a_1, a_2, w_1, w_2, w_0, \theta)$. Then objective function can be written as

$$f(z) = \sum_{i \in T_1} \left( \theta \left[ w_0 + w_1 a_1(x^i) + w_2 a_2(x^i) \right]_+ - y_i \right)^2 + \sum_{i \in T_2} \left( \theta \left[ w_0 + w_2 a_2(x^i) \right]_+ - y_i \right)^2$$
$$+ \sum_{i \in T_3} \left( \theta \left[ w_0 + w_1 a_1(x^i) \right]_+ - y_i \right)^2 + \sum_{i \in T_4} \left( \theta \left[ w_0 \right]_+ - y_i \right)^2$$

Now we can partition $T_1, T_2, T_3$ into $T_{1i}, T_{2i}, T_{3i}$ where $i = 1, 2$ respectively. For $T_{j1}$, ReLU terms in the objective, $f$, are constrained to be non-negative and for $T_{j2}$, ReLU terms are constrained to be non-positive, $j = 1, 2, 3$.

Observe that it suffices to check for $\theta, w_1, w_2 = \pm 1$. We will divide the optimization problem in two cases $w_0 \geq 0$ and $w_0 \leq 0$. In both cases, we will solve for convex program for all possible cases of $\theta, w_1, w_2 \in \{-1, 1\}$. So there are totally eight cases for each case of $w_0$. Since convex program is efficiently solvable so we get the global optimality of the training problem for 2-ReLU NN.

To claim global optimality, we need to enumerate all possible $Q_1/\overline{Q}_1$ and $Q_2/\overline{Q}_2$ which take $O(N^d)$ time each. Also total number of possible partitions is $O(N^d)$. So total $O(N^{2d})$ time to enumerate all possible combinations of $Q_1, Q_2$. For each enumerated combination, $Q_1, Q_2$, we again need to enumerate all possible partitions of $T_i$ which are $T_{i1}/T_{i2}, i = 1, 2, 3$. Using the same result Edelsbrunner et al. (1986), we know that this can be done in $O(|T_i|^d) = O(N^d)$ time again. We have to consider all possible combinations of partitions of $T_i$, hence we need $O(N^{3d})$ time to enumerate all possible combinations. Since we need to solve $O(N^{3d})$ convex programs for each choice of $Q_1, Q_2$ therefore we need to solve $O(N^{5d})$ convex programs. Each program can be solved in poly(N,d) time. Therefore overall this is poly(N) algorithm for a fixed $d$.

Now we give detail of convex program. Fix $\theta, w_1, w_2$ for some value in $\{-1, 1\}$. Then objective can be written as

$$f(z) = \sum_{i \in T_{11}} \left( \theta(w_0 + w_1 a_1(x^i) + w_2 a_2(x^i)) - y_i \right)^2 + \sum_{i \in T_{21}} \left( \theta(w_0 + w_2 a_2(x^i)) - y_i \right)^2$$
$$+ \sum_{i \in T_{31}} \left( \theta(w_0 + w_1 a_1(x^i)) - y_i \right)^2 + \sum_{i \in T_4} \left( \theta \left[ w_0 \right]_+ - y_i \right)^2 + \sum_{i \in T_{12} \cup T_{22} \cup T_{32}} (0 - y_i)^2$$

subject to constraints

$$
\begin{aligned}
a_1(x^i) &\geq 0, & i &\in Q_1 \\
a_1(x^i) &\leq 0, & i &\in \overline{Q}_1 \\
a_2(x^i) &\geq 0, & i &\in Q_2 \\
a_2(x^i) &\leq 0, & i &\in \overline{Q}_2 \\
w_0 + w_1 a_(x^i) + w_2 a_2(x^i) &\geq 0, & i &\in T_{11} \\
w_0 + w_1 a_(x^i) + w_2 a_2(x^i) &\leq 0, & i &\in T_{12} \\
w_0 + w_2 a_2(x^i) &\geq 0, & i &\in T_{21} \\
w_0 + w_2 a_2(x^i) &\leq 0, & i &\in T_{22} \\
w_0 + w_1 a_1(x^i) &\geq 0, & i &\in T_{31} \\
w_0 + w_1 a_1(x^i) &\leq 0, & i &\in T_{32}
\end{aligned}
$$

Moreover, we add constraint $w_0 \geq 0$ or $w_0 \leq 0$ and change the $[w_0]_+$ term in objective with $w_0$ or $0$ respectively. Every program has $2d + 3$ variables in $a_1, a_2, w_0$. Total number of constraints is at most $3N + 1$ so we conclude that this program can be solved in poly(N,d) time.

Since we enumerate over all possible partitions $Q_1, Q_2$ and $T_{i1}, i = 1, 2, 3$ hence we conclude that best solution out of all obtained solutions will be globally optimal.

## A.7 Proof of Theorem 3.4

We will show this for $0/1$ classification problem. But same ideas can be applied to a general output labels as well. Lets define $S_1 := \{i : y_i = 1\}$ and $S_0 := \{i : y_i = 0\}$.

Let $v \in \mathbb{R}^d$ be a random vector on a unit sphere, generated from a lebesgue measure. We will use the direction $v$ in each node of the first hidden layer of ReLU network. We claim that $v^T x^i, i \in [N]$ are distinct numbers with probability 1 with respect to the lebesgue measure.

Number of hyperplanes(or directions for normal vectors) in $\mathbb{R}^d$ that contain two or more data-points

$$\leq \sum_{i=2}^{\min\{d+1, N\}} \binom{N}{i} \leq 2^N.$$ Since $N$ is finite so there are finitely many directions. Probability of

selecting anyone of these $2^N$ directions from a lebesgue measure on unit sphere is zero. So we get the claim.

Now we claim that, at most $N$ nodes are needed in first hidden layer to generate an output $f(x) := \sum_j w_j [a_j]_+ (x)$ which hard-sorts the data. Once this is achieved, setting $w_0, \theta$ is trivial in the following way.

Suppose we hard-sorted the data such that if $i \in S_1$ then, we will make sure that $f(x^i) = 1$ otherwise $f(x^i) < 1$. This is clearly a hard-sorting solution. We will set $w_0 := -\max_{i \in S_0} f(x^i)$. Definition of $w_0$ and hard-sorting property of $f$ yields that $w_0 > -1$. We know that $w_0 + f(x^i) \leq 0$, for all $i \in S_0$ whereas $w_0 + f(x^i) = 1 + w_0 > 0$ for all $i \in S_1$. After setting $\theta = \frac{1}{1+w_0}$ it is easy to see that this choice of $w_0, \theta$ yields the final solution.

The only thing that remains to show is that we can obtain such solution $a_j, w_j$ in polynomial time such that $f$ hard-sorts the data and number of distinct nodes, or equivalently number of distinct functions $a_j$ required is at most $N$.

Suppose $z_i = v^T x^i$. By earlier claim, $\{z_i\}_{i \in [N]}$ are distinct numbers. Suppose $z_i$ are sorted in increasing order i.e. $z_{i+1} > z_i$ then, in one loop over the data, we can find required solution $\{a_j, w_j\}_{j \in [J]}$ and $J \leq N$. We present an exact procedure to find $a_j, w_j$ below.

Note that data is sorted according to $z_i$. We define $z_0 := z_1 - 1, y_0 = 0$. Current piece of piecewise linear function $f = \sum_j w_j [a_j]_+$ is stored in variable $g$. In particular, at each iteration $f(x^i) = g(x^i)$.

- **Initialization**: We skip all data points until we hit $i$ such that $y_i = 1$. Set $a_1(x) = \frac{1}{z_i - z_{i-1}}(v^T x - z_{i-1}), w_1 = 1$ and $g(x) = 0 + w_1 a_1(x)$. Note that $i$ could be 1 and we have defined $z_0, y_0$ so $a_1(x)$ is well defined. Without loss of generality, we assume that $y_1 = 1$. Current number of nodes is set in the variable $j$ initialized as $j = 1$.

- **Loop over data**: For i = 1 to N-1

  - **Case 1**: If $y_{i+1} = y_i = y_{i-1}$ then,
    skip the next steps and start with $i = i + 1$.

  - **Case 2**: If $y_{i+1} = y_i \neq y_{i-1}$ and $y_i = 1$ then,
    set $a_{j+1}(x) = g(x) - 1, w_{j+1} = -1, g(x) = 1, j = j + 1$.
    Skip the next steps an start with $i = i + 1$

  - **Case 3**: If $y_{i+1} = y_i \neq y_{i-1}$ and $y_i = 0$ then,
    skip the next steps and start with $i = i + 1$.

  - **Case 4**: If $y_{i+1} \neq y_i$ and $y_i = 0$ then,
    set $a_{j+1}(x) = \frac{1 - g(x^{i+1})}{z_{i+1} - z_i}(v^T x - z_i), w_{j+1} = 1, g(x) = g(x) + w_{j+1} a_{j+1}(x), j = j + 1$.
    Skip the next steps an start with $i = i + 1$.

  - **Case 5**: If $y_{i+1} \neq y_i$ and $y_i = 1$ then,
    set $a_{j+1}(x) = g(x) - 1 + v^T x - z_i, w_{j+1} = -1, g(x) = 1 + z_i - v^T x, j = j + 1$.
    Skip the next steps an start with $i = i + 1$.

We claim that in the algorithm presented above, for every iteration $i$, we compute new function $a_j$ in such a way that $\left[a_j\right]_+(x^k) = 0$ for all $k$ less than current iteration, say $i$. We also have that $\left[a_j\right]_+(x^i)$ is set in so that newly updated $f$ satisfies $f(x^i) \begin{cases} = 1 & \text{if } y_i = 1 \\ < 1 & \text{otherwise} \end{cases}$. Finally, newly computed $g$ is an affine function with normal vector which is 1)in direction of $-v$ if $y_i = 0$ and 2) in direction $v$ or $\mathbf{0}$ when $y_i = 1$. We will prove these facts by induction on $i$. We will call newly computed $g$ in each case as $g_{\text{new}}$ to distinguish between old and new function, $g$. Similarly after addition of a new node, the updated $f$ will called as $f_{\text{new}}$.

This is clearly true for $i = 1$ i.e. $a_1(x)$ and corresponding $g(x)$ defined in **Initialization** step satisfy all the conditions defined in inductive hypothesis. Suppose this is true for some $i$ and we need to show this for $i + 1$.

Case 1 and 3 are trivially true.

For case 2, we know that $f(x^k) \begin{cases} = 1 & \text{for } k \leq i \text{ and } y_i = 1 \\ < 1 & \text{for } k \leq i \text{ and } y_i = 0 \end{cases}$. So $a_j(x) = f(x) - 1 \leq 0$ for all $x \in \{x^1, \ldots, x^i\}$ as required. Moreover $g(x^{i+1}) > 1$ because $v^T x^i > v^T x^{i-1}$ and $g(x^i) > g(x^{i-1})$ then, $g$ must have normal in $+v$ direction. Therefore, we obtain $g(x^{i+1}) > g(x^i) = 1$ because $v^T x^{i+1} > v^T x^i$. Hence we conclude that newly added function, $a_j(x)$ satisfied $a_j(x^{i+1}) > 0$ so $f_{\text{new}}(x^k) = f(x) + w_j \left[a_j\right]_+ = \begin{cases} 1 & \text{for } k = i + 1 \\ f(x^k) & \text{for } k \leq i \end{cases}$. Finally at $x^{i+1}$ we have $g_{\text{new}}(x) = f(x) + w_j a_j(x) = 1$ so the normal is $\mathbf{0}$. This is valid since $y_{i+1} = 1$. This proves induction for case 2.

For case 4, since $y_i = 0$ so $g$ must be an affine function in $-v$ direction at $x^i$. Since $v^T x^{i+1} > v^T x^i$ so we obtain $g(x^{i+1}) < g(x^i) < 1$. So $\frac{1 - g(x^{i+1})}{z_{i+1} - z_i}$ is a positive scalar. Moreover, we have $v^T x^k \leq z_i$ for all $k \leq i$. So $\left[a_j\right]_+(x^k) = 0$ for all $k \leq i$. Also $a_j(x^{i+1}) = 1 - g(x^{i+1}) > 0$ therefore $f_{\text{new}}(x^{i+1}) = f(x^{i+1}) + \left[a_j\right]_+(x^{i+1}) = g(x^{i+1}) + \left[a_j\right]_+(x^{i+1}) = 1$. Finally, at $x^{i+1}$ we have that normal vector of $g_{\text{new}}$ must be either in direction $\pm v$ or $\mathbf{0}$ because normal of $a_j$ is in direction $+v$ and $g$ has normal either in direction $\pm v$ or is $\mathbf{0}$. Since $g(x^i) = f(x^i) = f_{\text{new}}(x^i) < f_{\text{new}}(x^{i+1}) = g_{\text{new}}(x^{i+1})$ therefore we conclude that normal must be in the direction $+v$. This is valid since $y_{i+1} = 1$. This proves induction for case 4.

For case 5, since $a_j(x) = g(x) - 1 + v^T x - z^i$, we can see that $a_j(x^k) \leq 0$ for $k \leq i$ because $v^T x^k \leq z_i$ and $f(x^k) \leq 1$. So we immediately get $f_{\text{new}}(x^k) = f(x^k)$ for all $k \leq i$. Since $y_i = 1$ so $g(x)$ has normal either in direction $+v$ or $\mathbf{0}$. In both cases, $g(x^{i+1}) \geq 1$. So we get that $a_j(x^{i+1}) \geq 0$. Hence $g_{\text{new}}(x^{i+1}) = g(x^{i+1}) + w_j a_j(x^{i+1}) = 1 + z_i - z_{i+1} < 1$. Finally, at $x^{i+1}$ we have that normal vector of $g_{\text{new}}$ must be either in direction $\pm v$ or $\mathbf{0}$ because of the form of $a_j$ and inductive hypothesis on $f$. Since $g_{\text{new}}(x^i) = 1 > g_{\text{new}}(x^{i+1})$ so normal must be in $-v$ direction which is valid since $y_{i+1} = 0$. This proves induction for case 5.

Since induction holds for all $i$ so $f(x) = \sum_{j \in J} w_j \left[a_j\right]_+(x)$ satisfies $f(x^k) = \begin{cases} 1 & \text{if } y_k = 1 \\ < 1 & \text{if } y_k = 0 \end{cases}$.

Moreover, in every iteration we add at most 1 node and there are total $N - 1$ iterations. So we can have at most $N$ nodes. (1 node is created in initialization step). Hence we conclude the proof.

Note that since $g, a_j$ are affine functions so there addition reduces to addition of vectors which can be done efficiently.

