# OpenReview forum: "Complexity of Training ReLU Neural Networks"
_ICLR.cc/2019/Conference_

### Official Review · AnonReviewer1 · 2018-11-02

**Rating:** 4
**Confidence:** 3

**Review:**

This paper shows that training of a 3 layer neural network with 2 hidden nodes in the first layer and one output node
is NP-complete. This is an extension of the result of Blum and Rivest'88. The original theorem was proved for
threshold activation units and the current paper proves the same result for ReLU activations. The authors do this
by reducing the 2-affine separability problem to that of fitting a neural network to data. The reduction is well
written and is clever. This is a reasonable contribution although it does not add significantly to the current state of the art.

---

### Official Review · AnonReviewer3 · 2018-11-03
**moderately interesting complexity result but perhaps the wrong venue**

**Rating:** 5
**Confidence:** 5

**Review:**

The main result of this work is to prove that a two-layer neural network with 2 hidden neurons in the first layer and 1 hidden neuron in the second layer is NP-hard to train when all activation functions are ReLU. Similar results with the hard-thresholding activation function were proved by Blum & Rivest in 1988. The main proof consists of a reduction from the 2-hyperplane separability problem which was known to be equivalent to NNs with hard-thresholding activation functions.

Quality: Moderate. Good solid formal results but nothing really surprising.

Clarity: The manuscript is mostly well-written, and the authors gave proper credit to prior works. The only minor issue is perhaps the abuse of the variable w in introduction and the same variable w (with an entirely different meaning) in the rest of the paper. It would make more sense to change the w's in introduction to d's.

Originality: This work is largely inspired by Blum & Rivest's work, and builds heavily on some previous work including Megiddo and Edelsbrunner et al. While there is certainly some novelty in extending prior work to the ReLU activation function, it is perhaps fair to say the originality is moderate.

Significance: While the technical construction seems plausible and correct, the real impact of the obtained results is perhaps rather limited. This is one of those papers that it is certainly nice to have all details worked out but none of the obtained results is really surprising or unexpected. While I do agree there is value in formally documenting the authors' results, this conference is perhaps not the right venue.

Other comments: The discussion on page 2 (related literature) seems odd. Wouldn't the results of Livni et al and Dasgupta et al already imply the NP-hardness of fully connected ReLU networks, in a way similar to how one obtains Corollary 3.2? If this is correct, then the contribution of this work is basically a refined complexity analysis where the ReLU network is shrunken to 2 layers with 3 nuerons?

I really wish the authors had tried to make their result more general, which in my opinion would make this paper more interesting and novel: can you extend the proof to a family of activation functions? It is certainly daunting to write separate papers to prove such a complexity result for every activation function... The authors also conveniently made the realizability assumption. What about the more interesting non-realizable case?

The construction to prove Theorem 3.4 bears some similarity to a related result in Zhang et al. The hard sorting part appears to be different.

---

### Official Review · AnonReviewer2 · 2018-11-04
**Complexity of training ReLU Neural Networks**

**Rating:** 3
**Confidence:** 5

**Review:**

This paper claims results showing ReLU networks (or a particular architecture for that) are NP-hard to learn. The authors claim that results that essentially show this (such as those by Livni et al.) are unsatisfactory as they only show this for ReLU networks that are fully connected. However, the authors fail to criticize their own paper for only showing this result for a network with 3 gates. For the same reason that the Livni et al. results don't imply anything for fully connected networks, these results don't imply anything for larger networks. Conceivably certain gadgets could be created to ensure that the larger networks are essentially forced to ignore the rest of the gates. This line of research isn't terribly interesting and furthermore the paper is not particularly well written.

For learning ReLUs, it is already known (assuming conjectures based on hardness of improper PAC learning) that functions that can be represented as a single hidden layer ReLU network cannot be learned even using a much larger network in polynomial time (see for instance the Livni et al. paper, etc.). Proving NP-hardness results for proper isn't as useful as they usually are very restricted in terms of architectures the learning algorithm is allowed to use. However, if they do want to show such results, I think the NP-hardness of learning 2-term DNF formulas will be a much easier starting point.

Also, I think there is a flaw in the proof of Lemma 4.1. The function f *cannot* be represented by the networks the authors claim to use. In particular the 1/\eta outside the max(0, x) term is not acceptable.

---

### Meta-Review · Area_Chair1 · 2018-12-13
**Restricted theoretical result**

**Confidence:** 4
**Recommendation:** Reject

**Metareview:**

Dear authors,

All reviewers agreed that, while the problem considered was of interest, the theoretical result presented in this work was of too limited scope to be of interest for the ICLR audience.

Based on their comments, you might want to consider a more theoretically-oriented venue for such a submission.